# Combination of Inorganic Nitrogen and Organic Soil Amendment Improves Nitrogen Use Efficiency While Reducing Nitrogen Runoff

Ian Phillips [1], Chanyarat Paungfoo-Lonhienne [2,*], Iman Tahmasbian [1], Benjamin Hunter [1], Brianna Smith [1], David Mayer [1] and Matthew Redding [1]

1   Department of Agriculture and Fisheries, Queensland Government, P.O. Box 102, Toowoomba, QLD 4350, Australia; Ian.Phillips@daf.qld.gov.au (I.P.); Iman.Tahmasbian@daf.qld.gov.au (I.T.); benchunter1@gmail.com (B.H.); Brianna.Smith@daf.qld.gov.au (B.S.); David.Mayer@daf.qld.gov.au (D.M.); Matthew.Redding@daf.qld.gov.au (M.R.)
2   School of Agriculture and Food Science, University of Queensland, St. Lucia, QLD 4072, Australia
*   Correspondence: chanyarat@uq.edu.au; Tel.: +61-04-4878-5658

**Abstract:** Improved nitrogen fertiliser management and increased nitrogen use efficiency (NUE) can be achieved by synchronising nitrogen (N) availability with plant uptake requirements. Organic materials in conjunction with inorganic fertilisers provide a strategy for supplying plant-available N over the growing season and reducing N loss. This study investigated whether a combined application of inorganic N with an organic soil amendment could improve nitrogen use efficiency by reducing N loss in runoff. Nitrogen runoff from a ryegrass (*Lolium multiflorum*) cover was investigated using a rainfall simulator. Nitrogen was applied at low, medium and high (50, 75 and 100 kg/ha) rates as either $(NH_4)_2SO_4$ or in combination with a poultry manure-based organic material. We showed that the NUE in the combination (58–75%) was two-fold greater than in $(NH_4)_2SO_4$ (24–42%). Furthermore, this combination also resulted in a two-fold lower N runoff compared with the inorganic fertiliser alone. This effect was attributed to the slower rate of N release from the organic amendment relative to the inorganic fertiliser. Here, we demonstrated that the combined use of inorganic and organic N substrates can reduce nutrient losses in surface runoff due to a better synchronisation of N availability with plant uptake requirements.

**Keywords:** rainfall simulator; nutrient runoff; ammonium; nitrate; nitrogen use efficiency

## 1. Introduction

The efficient use of fertilisers relies on optimising the time of application to meet crop nutrient requirements, and minimising nutrient losses through processes such as leaching and surface water runoff. With the intensification of Australian agriculture in response to increased global food requirements, the use of nitrogen-based fertilisers has concomitantly increased dramatically [1–4]. However, <50% of applied nitrogen (N) is effectively utilised by the growing plant in many cropping situations, and this unutilised N represents a significant economic loss and can pose a high risk of environmental pollution [4–8].

At the same time, the need for developing more agricultural land to meet global food requirements has seen an increased research focus on the role of sandy soils in cropping systems, and on identifying management strategies to overcome cropping constraints [9]. Despite sands being perceived as highly leachable, this characteristic is of less importance in shallow sands underlain by low-permeability layers due to increasing clay content and/or mechanical impedance/compaction [10]. Therefore, any decline in infiltrability or deep drainage can encourage other water loss mechanisms, such as runoff [11].

There has been renewed interest in the use of organic fertilisers for supplying plant nutrients [12,13], and as a feedback mechanism for improving soil health [14]. The term

"organic fertiliser" covers a wide range of substrates including manures, composts, and plant stubble and root residues [13,15,16], and these substrates vary considerably in nutrient content in both form and concentration [17]. However, there are issues relating to the slow rate of N release from organic substrates. The subsequent slow kinetics of mineralisation and nitrification rates have brought into question the sole reliance on organic fertilisers for satisfying the plant nutrient requirements [18,19]. Paungfoo-Lonhienne et al. (2019) [20] reported less biomass for kikuyu grass (*Pennisetum clandestinum*) grown in a poultry manure-based organic fertiliser (CropUp$^{TM}$) compared with an inorganic fertiliser (urea). These researchers suggested that the rate of N supply (including resident $NH_4$ and $NO_3$, and mineralised N) from the organic substrate alone was insufficient to meet the N demand of the growing plant. To address this N limitation, plants were grown in soil receiving a combination of 50% urea and 50% CropUp$^{TM}$. This combination significantly decreased mineral N in leachates compared to urea alone. However, in contrast to leachate N, the soil mineral N levels for the combined (organic + urea N sources) and urea-only treatments were similar, indicating that little N mineralisation from the organic substrate had occurred. Although organic fertilisers are capable of supplying N for plant use, factors such as the organic substrate and soil chemical and physical characteristics, as well as microbial diversity and functionality, play key roles in the mineralisation and nutrient release rates from these organic materials [13,15,16,19].

The retention of inorganic and organic forms of N closer to the soil surface due to reduced leaching and mineralisation rates can expose this nutrient to loss mechanisms such as wind and water erosion, depending on climatic and soil physical conditions. Soluble forms of both organic and inorganic N are the most prone to loss, particularly in surface runoff. As global demand for agricultural land increases, sandy soils with low nutrient and water retention properties are expected to play an increasing role in food production. The combined use of inorganic and organic sources of N may provide one strategy to minimise nutrient loss and maximise N use efficiency. Although the leaching of N supplied as combined inorganic/organic substrates has been investigated [20], losses from runoff, particularly for sandy soil, are scarce [21–23].

The aim of this study was to investigate the suitability of a combined inorganic and organic fertiliser to improve nitrogen use efficiency (NUE) in a sand. Specifically, this study targeted (1) N losses in surface runoff from well-controlled rainfall simulation experiments for a combined (organic + inorganic sources) and an inorganic N-based fertiliser, and (2) laboratory-scale N release from inorganic and organic N sources.

## 2. Materials and Methods

### 2.1. Rainfall Simulation Trial

Rainfall runoff trials were undertaken using a nutritionally deficient (low total and available N) coarse-textured sand (Table 1). This material was selected as the growing medium to reflect the potential N runoff loss from a sandy-textured soil, to minimise any potential extraneous N sources for plant uptake (e.g., mineralisation of resident organic N), and to minimise suspended colloid concentrations that could preferentially transport fertiliser N in runoff via ion adsorption. This coarse sandy material therefore maximised the potential N losses in runoff as a means of evaluating the benefits of organic versus inorganic sources of N in maximising plant uptake.

**Table 1.** Selected properties of the sand. EC = electrical conductivity, ECEC = effective cation exchange capacity.

| Parameter | Sand |
|---|---|
| $pH_{1:5}$ | 6.15 |
| $EC_{1:5}$ (dS/m) | 0.02 |
| Total Carbon (%) | 0.22 |
| Total Nitrogen (%) | 0.022 |
| Organic Carbon (%) | 0.22 |
| $NH_4$-N (mg/kg) | 3 |
| $NO_3$-N (mg/kg) | <2 |
| Exchangeable Cations ($cmol_+$/kg) | |
| Ca | 0.634 |
| Mg | 0.373 |
| K | 0.114 |
| Na | <0.080 |
| ECEC | 1.12 |
| Particle Size Distribution (%) | |
| Coarse Sand | 62.5 |
| Fine Sand | 30.2 |
| Silt | 6.3 |
| Clay | 3.6 |
| Texture | coarse sand |

The sand (total sand ≈ 93%) was slightly acidic (pH 6.15), non-saline, and contained extremely low organic carbon (organic C = 0.22%) and nitrogen (total N = 0.022%) concentrations. The effective cation exchange capacity (ECEC = 1.12 $cmol_+$/kg) was very low, with calcium being the dominant exchangeable cation, along with lesser concentrations of magnesium and potassium.

Nitrogen was added as either a poultry-based organic fertiliser (CropUp$^{TM}$) or as inorganic ammonium sulphate (($NH_4$)$_2SO_4$). CropUp$^{TM}$ is a mixture of composted manure, molasses, humates and natural minerals (14.5% zeolite), and is slightly soluble in water (Sustainable Organic Solutions Pty Ltd., Brisbane, QLD, Australia—Safety Data Sheet). CropUp$^{TM}$ contains 3.07% N (as total N; LECO CN Dumas analyser, St Joseph, MI, USA), 23.25% C (as total C; LECO CN Dumas analyser), a C:N ratio of 7.56, 4544.49 mg/kg of available $NH_4^+$ (2M KCl-extractable; [24]; Method 7C2), 40.67 mg/kg of available $NO_3^-$ (2M KCl-extractable; [24]; Method 7C2), a pH of 8.41 (1:5 CropUp$^{TM}$:18.2 MΩ deionised water) and an EC of 9.18 dS/m (1:5 CropUp$^{TM}$:18.2 MΩ deionised water).

Ammonium sulphate, rather than urea [20], was used as the inorganic N source to minimise the potential for $NH_3$ volatilisation. Inorganic N was applied at rates of 0 (Control), 50 (ASLow), 75 (ASMedium) and 100 (ASHigh) kg N/ha as ($NH_4$)$_2SO_4$ (Table 2), which reflects the application rates used by previous workers [20]. The poultry manure-based organic material CropUp$^{TM}$ was applied to achieve N rates of 25 (CULow), 37.5 (CUMedium) and 50 (CUHigh) kg/ha, and was supplemented with ($NH_4$)$_2SO_4$ (25, 37.5 and 50 kg N/ha, respectively) to match the amount of N added in inorganic N treatments, and to ensure that N availability was not limited during the plant establishment and growth phases.

**Table 2.** Treatments used in the rainfall simulation experiment. "AS" indicates inorganic N source $(NH_4)_2SO_4$ and "CU" indicates combined $(NH_4)_2SO_4$ + CropUp[TM].

| Treatment | $(NH_4)_2SO_4$ (kg/ha) | CropUp (kg/ha) | $(NH_4)_2SO_4$ (g/kg Soil) | CropUp (g/kg Soil) |
|---|---|---|---|---|
| Control | 0 | 0 | 0.000 | 0.000 |
| ASLow | 50 | 0 | 0.546 | 0.000 |
| CULow | 25 | 25 | 0.273 | 2.593 |
| ASMedium | 75 | 0 | 0.819 | 0.000 |
| CUMedium | 37.5 | 37.5 | 0.409 | 3.890 |
| ASHigh | 100 | 0 | 1.092 | 0.000 |
| CUHigh | 50 | 50 | 0.546 | 5.187 |

The sand was packed into stainless steel trays (1045 × 457 × 40 mm; *n* = 3) to achieve an approximate bulk density ($\rho_b$) of 1110 kg/m$^3$. The sand was initially packed to a height of 30 mm, and the various treatments (Table 2) were uniformly surface applied. The treatments were covered with an additional 7 mm of sand, and lightly compacted to produce a relatively uniform surface. Ryegrass (*Lolium multiflorum*) seed was spread across the sand surface at a rate, equivalent on a surface area basis, of 200 kg/ha. The grass seed was covered with an additional 3 mm of sand, and the soil tray was slowly moistened with water. Water was applied to achieve an approximate gravimetric water content ($\theta_g$) corresponding to 60% of field capacity ($\theta_{fc}$). The soil trays were maintained at this moisture content for a period of 42 days prior to undertaking rainfall simulation trials.

To minimise the likelihood of nutrient deficiencies limiting the ryegrass growth, each tray received a basal nutrient application equivalent, on a surface area basis, to 20 kg P/ha, 100 kg K/ha, 28 kg Mg/ha, 70 kg S/ha, 0.43 kg Cu/ha, 0.84 kg Zn/ha, 7.7 kg Mn/ha, 0.97 kg B/ha, 0.33 kg Mo/ha and 30 kg Ca/ha.

Nutrient runoff from the treated soil trays was generated using a rainfall simulator built in accordance with published specifications [25], using a similar procedure described by [26]. The rainfall simulator was positioned centrally over two flumes, and the simulation was conducted at a nozzle pressure of 28 kPa (the design pressure required to deliver a rainfall intensity of 70 mm/h) over a runoff period of 20 min.

Prior to commencing each simulation run, the soil surface was photographed as a measure of grass cover, as this parameter can influence surface runoff rates, and hence the nutrient loading of the runoff water. The photographs were taken orthogonally to the soil surface under uniform light conditions. The software was written in Python using the OpenCV library (Python Software Foundation, 2019) to enable a uniform set of pixel colours (defined by hue, saturation, and value, the standard HSV digital colour space) to be selected as either soil or plant matter across all the images after gamma-balancing each image. Plant coverage was calculated from the ratio of background to plant pixel counts and validated automatically via the total percentage of area covered by soil and the percentage of area covered by plant pixels. The set of pixel characteristics selected was modified to minimise discrepancies across these three methods iteratively, and then applied uniformly across all images.

After being placed in the flume, each treatment tray was manually wet up to saturation prior to commencing rainfall. A sample of rainfall water was collected for analysis as outlined below for the runoff samples. A composite sample of the water exiting the flume of each treatment was automatically collected (WS750 water sampler, Global Water Instrumentation Inc., College Station, TX, USA) at the commencement of flow and subsequently every 5 min (100 mL aliquot composited for each 5 min collection event). The height of the water at the flume discharge point was measured at each 5 min sampling period, and the volume flow rate (V) was calculated using [27]:

$$V = 341 \times H^{2.31} \qquad (1)$$

where V = volume flow rate (L/s) and H = head of water (m).

At the end of each simulation period, eight cores (internal diameter = 110 mm) were removed from the treatment tray. The cumulative area of the eight cores was, on a surface area basis, approximately 8% of the treatment area. The composited sand core materials (soil + grass) were thoroughly mixed and then stored at <4 °C prior to analysis.

The runoff water, soil and plant materials were analysed as follows. An aliquot of each sample of runoff water was initially filtered (<0.45 μm), and samples of unfiltered and filtered water were analysed for total N (APHA 5310B). The filtered samples were also analysed colorimetrically for $NH_4^+$ [28] and $NO_3^- + NO_2^-$ using a modified Griess method [29] with a microtiter plate reader (BioTek EPOCH$^2$ Microplate Reader) at a wavelength of 625 and 540 nm, respectively. Runoff nutrient and particulate concentrations were converted to mass loss to account for small variations in flow rate between the two flumes using Equation (1).

Sub-samples of soil (*n* = 3) were analysed for mineral N (2M KCl extractable; [24]; Method 7C2). Plant material (above- and below-ground material) was separated from the soil by washing with Milli-Q deionised water. The retained plant material was oven-dried at 60 °C, weighed to estimate the dry matter (DM for whole plant biomass), and ground prior to analysis for total N and total C by high-temperature combustion (LECO CN Analyser), and for aluminium, boron, calcium, copper, iron, potassium, magnesium, manganese, sodium, phosphorus, sulphur and zinc by nitric acid digestion and ICPOES.

*2.2. Nutrient Release Trial*

This experiment was a laboratory-scale leaching study to investigate N mineralisation kinetics using coarse-textured sand (Table 1). Nitrogen was added at a rate of 291 kg/ha as either CropUp$^{TM}$ (equivalent to 8.675 mg/g soil) or as ammonium chloride ($NH_4Cl$) (equivalent to 0.9274 mg/g soil). This rate is higher than that used in the runoff trial (50–100 kg N/ha) to ensure that effects of dissolution and transformation were measurable. Sand with no N applied served as a control. Leaching was undertaken in 50 mL polypropylene centrifuge tubes with a small hole (5 mm diameter) drilled through the base to allow for drainage of the leachate. A disc of Whatman glass fibre paper was placed inside the tube to cover the hole so as to avoid soil loss during leaching.

Air-dried (<2 mm) sand (equivalent to 40 g on an oven-dry weight basis) was placed into the respective centrifuge tube. Initially, approximately 80–85% of the total soil mass was added. The respective treatments were then added to the sand surface, with CropUp$^{TM}$ as solid material and $NH_4Cl$ in 1 mL of solution. The remaining 15–20% of the sand was then placed above the soil/treatment interface. The sand was lightly compacted by dropping the tube 10–15 times vertically from a 2 cm height. This packing procedure resulted in the treatments residing ≈10 mm below the sand surface.

Each sand column was leached with the equivalent of 1.5 pore volumes of 0.005 M $CaCl_2$ (≈12.7 mL) on days 1, 3, 10, 17, 24, 33 and 42 of the study. Dilute $CaCl_2$ was chosen, rather than water, as the leaching solution to simulate the soil solution's ionic composition and ionic strength, and to minimise the likelihood of, albeit a low amount of, mobilised clay clogging the filter disc.

Leaching occurred over a 2–3 h period (under gravity/free drainage), and the leachates were collected in 70 mL polypropylene containers. After drainage ceased, each leaching tube was placed under a slight vacuum for 3 to 5 s to remove excessive solution from the base of the tube and to avoid generating anaerobic conditions. The drainage collected under vacuum was added to the gravity drainage, and the volume of the leachate was estimated by weighing.

All the leachates were filtered (<0.45 μm cellulose acetate filter membranes) and stored frozen before being analysed colormetrically for $NO_x$ and $NH_4$ using a microtiter plate reader (BioTek EPOCH$^2$ Microplate Reader) at a wavelength of 625 and 540 nm, respectively. Following each leaching event, the tubes were re-capped (lids remained loose to maintain aerobic conditions) and incubated at 25 °C.

After the final leaching (day 42), the remaining available mineral N was extracted from the sand using 2 M KCl (2 h end-over-end shaking at a 1:5 soil-to-solution ratio). After extraction, the supernatant was removed, centrifuged (10,000 rpm for 20 min), filtered (<0.45 µm) and stored frozen prior to $NO_x$ and $NH_4$ analysis. The concentrations of $NO_x$ and $NH_4$ were corrected by subtracting the amount of each ion retained in the entrained solution.

### 2.3. Statistical Analysis

General linear mixed models (GLMs) were used to analyse the rainfall simulator data, using restricted maximum likelihood (REML) in GenStat (2018). Residual plots were used to confirm the assumptions of homogeneous variances and low skewness, with flow rate as the standardising covariate. The treatments were initially analysed as 13 discrete levels, and subsequently as the factorial structure of N-rates by N-sources. A post hoc comparison between the adjusted means was performed using protected least significant difference testing at a significance level of 5% ($p < 0.05$). Nitrogen release data were analysed using analysis of variance (general linear model; *Statistix* version 10). The significant differences among the main treatments were separated by LSD ($p < 0.05$). The relationships between cumulative leachate $NH_4$ and $NO_x$ with time were adequately described ($r^2$ for $NH_4$ = 0.85–0.95 and $r^2$ for $NO_x$ = 0.75–0.89) by an equation of the form:

$$N_i = k \times \ln t + N_0 \tag{2}$$

where $N_i$ = leachate N (mg $NH_4$ or $NO_x$) at time $i$, $t$ = time (days), $N_0$ = leachate N (mg $NH_4$ or $NO_x$) at time 0 and $k$ = first-order rate coefficient ($d^{-1}$).

### 3. Results

A rainfall simulation trial was undertaken to evaluate the impacts of combined inorganic and organic fertiliser on nitrogen runoff losses compared with conventional inorganic fertiliser (($NH_4)_2SO_4$) alone. A nitrogen release experiment was concurrently undertaken to investigate N mineralisation kinetics.

### 3.1. Nitrogen Release from Organic and Inorganic Sources

A leaching experiment was undertaken to investigate the N mineralisation kinetics in the coarse-textured sand used in the rainfall simulation trial. The results show that negligible $NH_4$ and $NO_x$ were leached over the 42-day period in the control treatment (no N applied), confirming the very low mineral N status of the sand (Figure 1). The leachate from the CropUp$^{TM}$ treatment contained very low amounts of $NH_4$ (<0.7 mg), which were often not significantly ($p < 0.05$) different from that found in the control. For example, leachate $NH_4$ initially increased from 0.09 mg on day 1 to 0.67 mg on day 3, after which leachate $NH_4$ decreased steadily to background (control) levels. Leachate $NO_x$ from CropUp$^{TM}$ remained very low over the initial 3 days, but increased to 0.17 mg by day 10, and 0.28 mg by day 17. By day 24, the amount of $NO_x$ in the leachate was not significantly different from that found in the control.

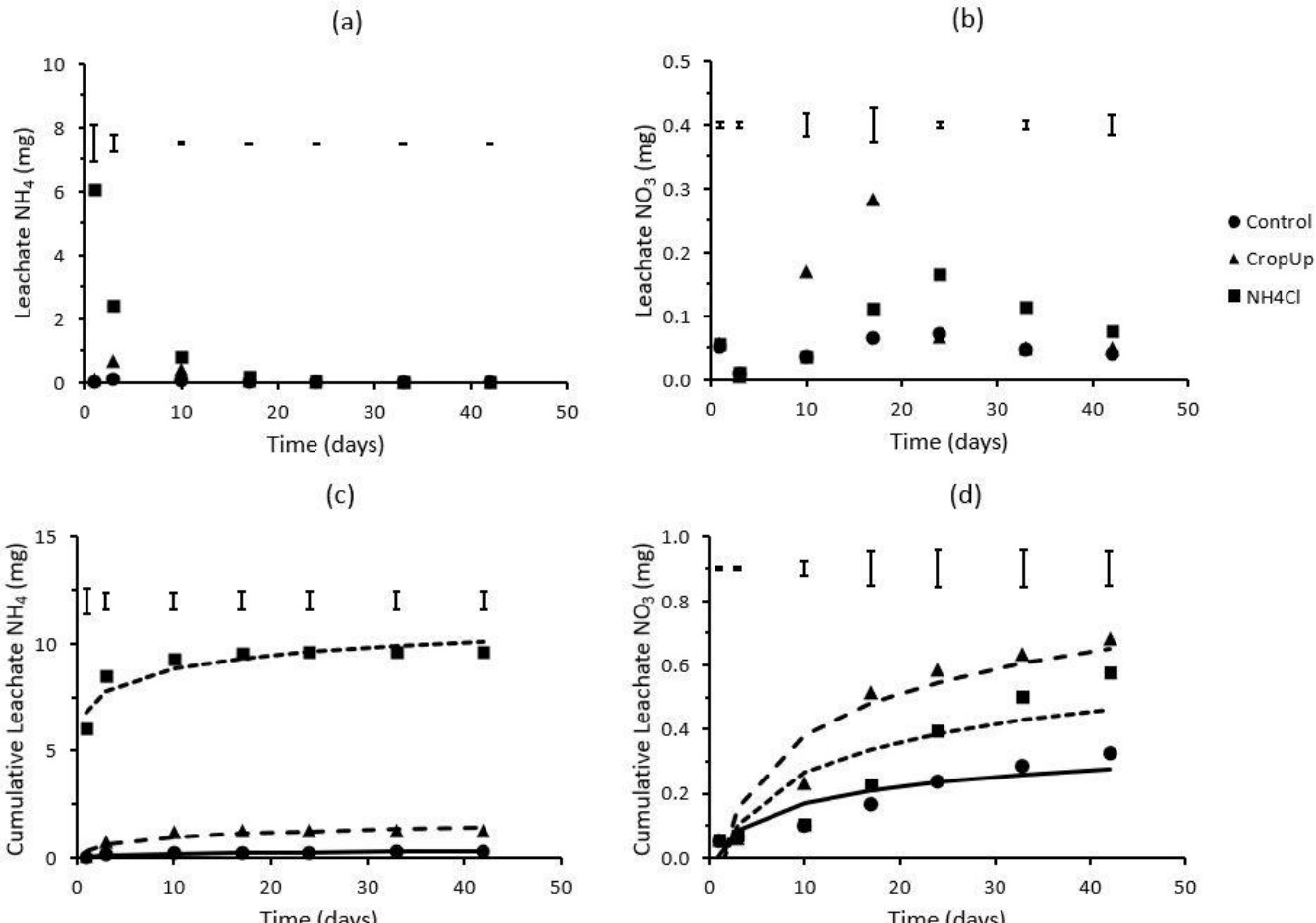

**Figure 1.** Leachate (**a**) $NH_4$ and (**b**) $NO_x$, and cumulative leachate (**c**) $NH_4$ and (**d**) $NO_x$ as a function of time. Bars represent LSD values ($p < 0.05$). The solid and dashed lines in (**c**) and (**d**) were predicted using Equation (2).

The majority (>95%) of the mineral N leached from the $NH_4Cl$ treatment was in the $NH_4$ form with little conversion to $NO_x$ (Figure 1). Of this $NH_4$, ≈96% leached from the sand column within the initial 10 days of the study (corresponding to 2–3 pore volumes of leachate). Levels of $NO_x$ ranging from 0.11 to 0.16 mg were measured between day 17 and day 24 (corresponding to 3–4 pore volumes), but declined to background levels with increased leaching. The distributions of $NH_4$ and $NO_x$ in the leachate showed that the majority of $NH_4$ was readily leached from the sand, but a small amount of residual $NH_4$ was nitrified during the later stage of the study.

Over the 42-day incubation period, 0.26, 1.26 and 9.64 mg of $NH_4$ and 0.33, 0.68 and 0.58 of $NO_x$ leached from the control, CropUp$^{TM}$ and $NH_4Cl$ treatments, respectively (Figure 1c,d). The amount of cumulative mineral N ($NH_4 + NO_x$) that leached from the control, CropUp$^{TM}$ and $NH_4Cl$ treatments was 0.59, 1.94 and 10.22 mg, respectively (Table 3). The ammonium and $NO_3$ release rate coefficients (Equation (2)) for the control, CropUp$^{TM}$ and $NH_4Cl$ treatments were 0.062, 0.302 and 0.884 d$^{-1}$, and 0.073, 0.187 and 0.137 d$^{-1}$, respectively.

After 42 days, much of the (2M KCl-extractable) mineral N was present as $NH_4$, with negligible $NO_x$, irrespective of the treatment (Figure 2). Mean $NH_4$ concentrations ranged from 12.49 mg/kg in the control to 13.60 mg/kg in the $NH_4Cl$ treatment, but these were not significantly different from each other. Mean $NO_x$ concentrations ranged from ≈0 mg/kg in the control and $NH_4Cl$ treatments to 1.05 mg/kg in the CropUp$^{TM}$ treatment, with the majority of extracted $NO_x$ present in the pore water.

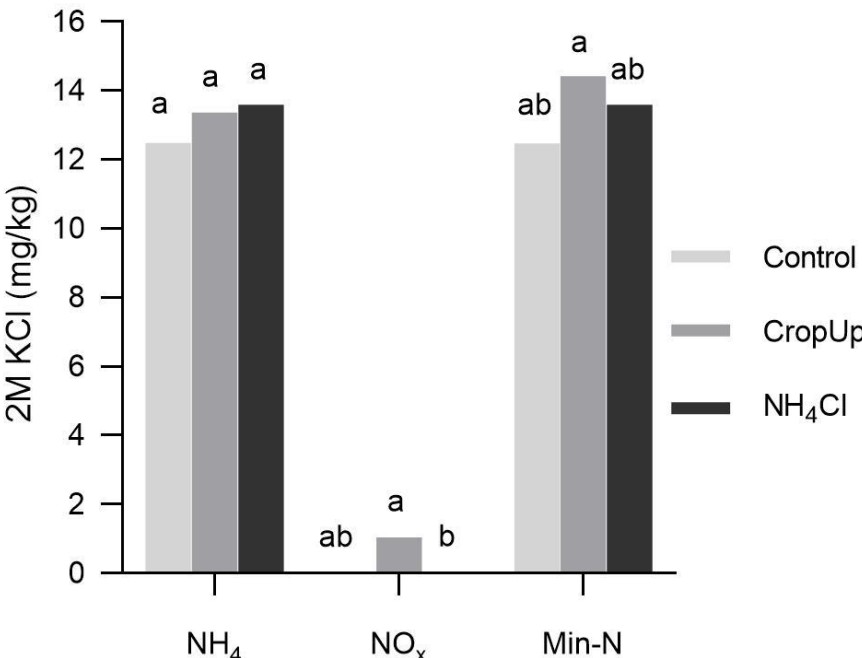

**Figure 2.** Mean 2M KCl-extractable $NH_4$ and $NO_x$, and mineral N ($NH_4$ + $NO_x$) concentrations for the control, CropUp[TM] and $NH_4Cl$ treatments. Parameters with the same letter are not significantly different at $p < 0.05$ from each other.

The amounts of mineral N (i.e., $NH_4$ + $NO_x$) recovered for the control, CropUp[TM] and $NH_4Cl$ treatments were 1.14, 2.58 and 10.84 mg, respectively (Table 3). Of these amounts, 2M KCl-extractable $NH_4$ was ≈0.5 mg for all treatments, suggesting that little to no $NH_4$ from either CropUp[TM] or $NH_4Cl$ was retained by the sand. This small quantity of $NH_4$ may represent a background concentration that cannot be displaced from the cation exchange sites by K-$NH_4$ or Ca-$NH_4$ exchange.

### 3.2. Rainfall Runoff Trial

The plant dry matter displayed a strong positive, linear relationship with the plant cover (Figure 3; *plant cover* = 0.626*DM* + 1.95; $r^2$ = 0.94). The treatments receiving inorganic N fertiliser in the form of $(NH_4)_2SO_4$ (ASLow, ASMedium and ASHigh) contained significantly lower plant dry matter and percentage cover than the treatments receiving the combined ($(NH_4)_2SO_4$ + CropUp[TM]) fertiliser (CULow, CUMedium and CUHigh). The ammonium sulphate treatments produced a plant cover of approximately 40–45%, while the combined $(NH_4)_2SO_4$ + CropUp[TM] produced plant covers ranging from 70 to 90% (Figure 3). The lower plant cover was consistent with the relatively poor ryegrass (*Lolium multiflorum*) seed germination observed during the 42-day growth stage.

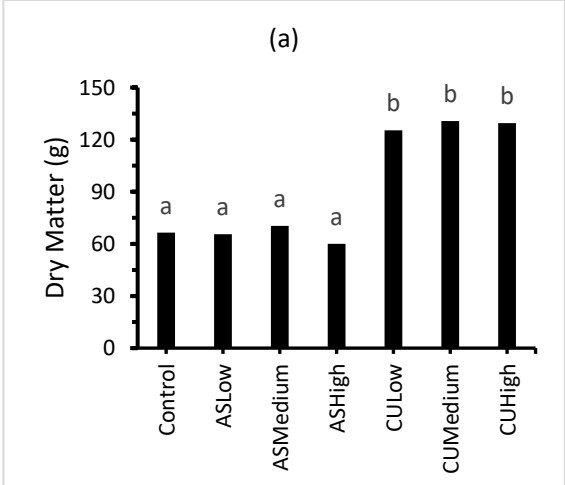

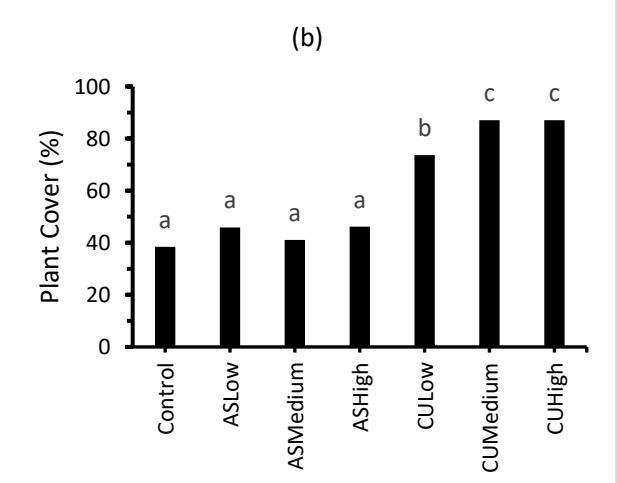

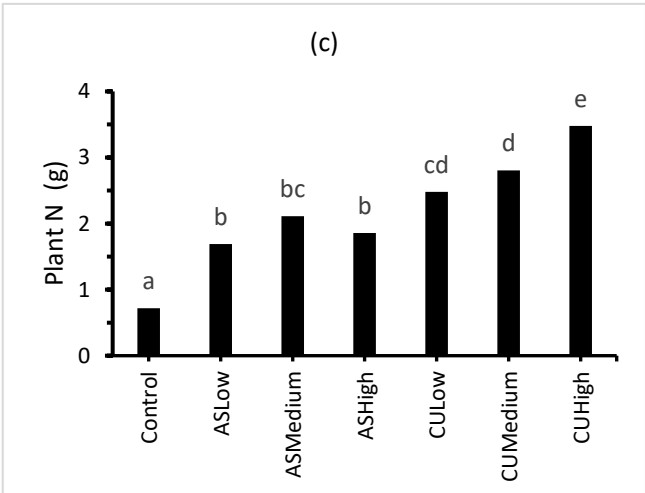

**Figure 3.** Effect of treatment on: (**a**) percentage plant cover; (**b**) plant biomass; and (**c**) plant total N. Bars represent the mean of three replicates (*n* = 3) and bars designated by the same letter are not significantly different at *p* < 0.05.

The ryegrass treatments (CULow, CUMedium and CUHigh) contained significantly (*p* < 0.05) more plant C and plant N than ASLow, ASMedium and ASHigh (Table 4 and Figure 3). The plant macronutrient (P, K, Ca, Mg and S) and micronutrient (Al, B, Fe, Mn, Na and Zn) content was significantly higher for the ryegrass grown in the presence of CropUp$^{TM}$ compared with the control and $(NH_4)_2SO_4$ treatments (Table 4).

**Table 3.** Distribution of $NH_4$, $NO_x$ and mineral N in the soil and leachate, and percentage recovery of added N.

| Treatment | 2M KCl (mg) | | Leachate (mg) | | Sum (mg) | | Min-N (mg) | Increase in (mg) | | | % Recovery | | |
|---|---|---|---|---|---|---|---|---|---|---|---|---|---|
| | $NH_4$ | $NO_x$ | $NH_4$ | $NO_x$ | $NH_4$ | $NO_x$ | | $NH_4$ | $NO_x$ | Min-N | $NH_4$ | $NO_x$ | Min-N |
| Control | 0.511 | 0.037 | 0.258 | 0.329 | 0.770 | 0.366 | 1.136 | | | | | | |
| CropUp$^{TM}$ | 0.547 | 0.088 | 1.260 | 0.683 | 1.808 | 0.771 | 2.579 | $1.038 \pm 0.106$ | $0.405 \pm 0.145$ | $1.443 \pm 0.155$ | 10.684 | 4.171 | 14.855 |
| $NH_4Cl$ | 0.574 | 0.049 | 9.641 | 0.577 | 10.215 | 0.626 | 10.841 | $9.446 \pm 0.937$ | $0.259 \pm 0.012$ | $9.705 \pm 0.939$ | 97.220 | 2.672 | 99.892 |

**Table 4.** Nutrient content in the grass cover (Lolium multiflorum) after the 42-day growing period for each treatment. Parameters designated by the same letter are not significantly different at $p < 0.05$.

| Treatment | TC | TN | Al | B | Ca | Cu | Fe | K | Mg | Mn | Na | P | S | Zn | NUE * |
|---|---|---|---|---|---|---|---|---|---|---|---|---|---|---|---|
| | (g) | | | | | | | | | | | | | | (%) |
| Control | 23565 b | 691 e | 203.5 b | 0.40 c | 189.69 c | 6.04 a | 213.5 b | 208.8 b | 84.1 b | 7.08 b | 35.97 b | 58.61 d | 70.0 f | 78.9 d | - |
| ASLow | 26643 b | 1690 d | 236.5 b | 0.39 c | 143.86 cd | 4.23 b | 200.2 b | 313.9 b | 70.7 bc | 6.60 b | 29.63 bc | 74.69 cd | 157.6 e | 206.7 c | 42 |
| CULow | 46982 a | 2480 bc | 392.7 a | 0.84 b | 432.60 b | 5.99 a | 374.8 a | 721.2 a | 150.8 a | 13.59 a | 58.50 a | 164.79 b | 225.4 bc | 296.9 b | 75 |
| ASMedium | 28696 b | 2112 cd | 236.0 b | 0.40 c | 137.39 cd | 4.31 b | 203.7 b | 418.8 ab | 72.9 bc | 7.04 b | 31.90 bc | 97.35 c | 197.6 cd | 317.6 ab | 40 |
| CUMedium | 48472 a | 2805 bc | 438.0 a | 0.85 b | 418.99 b | 6.12 a | 400.6 a | 462.5 ab | 153.6 a | 13.70 a | 60.23 a | 181.34 b | 254.4 b | 325.3 ab | 59 |
| ASHigh | 24442 b | 1856 d | 201.6 b | 0.34 c | 119.09 d | 3.42 b | 189.2 b | 426.6 ab | 62.5 c | 6.33 b | 27.04 c | 86.12 cd | 172.1 de | 284.8 b | 24 |
| CUHigh | 50864 a | 3477 a | 382.9 a | 1.00 a | 496.22 a | 6.24 a | 376.1 a | 628.1 b | 164.0 a | 15.22 a | 58.72 a | 220.23 a | 317.4 a | 380.4 a | 58 |

* NUE = nitrogen use efficiency calculated as: ((Plant N per treatment − Plant N in control) ÷ N added in Fertiliser) × 100 [30–32].

The nitrogen use efficiency (NUE) was estimated as the proportion of applied N taken up by the ryegrass (Table 4). The NUE for the $(NH_4)_2SO_4$-only treatments ranged from 58% to 75% in the $(NH_4)_2SO_4$ + CropUp[TM] treatments, and was two-fold greater than for the $(NH_4)_2SO_4$) treatment (24–42%).

The masses of total N, $NH_4$, $NO_x$ and mineral N ($NH_4$ + $NO_x$) in the runoff from the Control, ASLow, ASMedium, ASHigh, CULow, CUMedium and CUHigh treatments are presented in Figure 4. The masses of runoff total N, $NH_4$ and mineral N for the inorganic N fertiliser ($(NH_4)_2SO_4$) were higher ($p < 0.05$) than for the $(NH_4)_2SO_4$ + CropUp[TM] treatments. For example, $\geq 200$ mg of total N was collected for the inorganic N treatments compared with $\approx 100$ mg for the $(NH_4)_2SO_4$ + CropUp[TM] treatments, which represented a two-fold reduction in N runoff. The runoff $NO_3$ content was very low for all treatments.

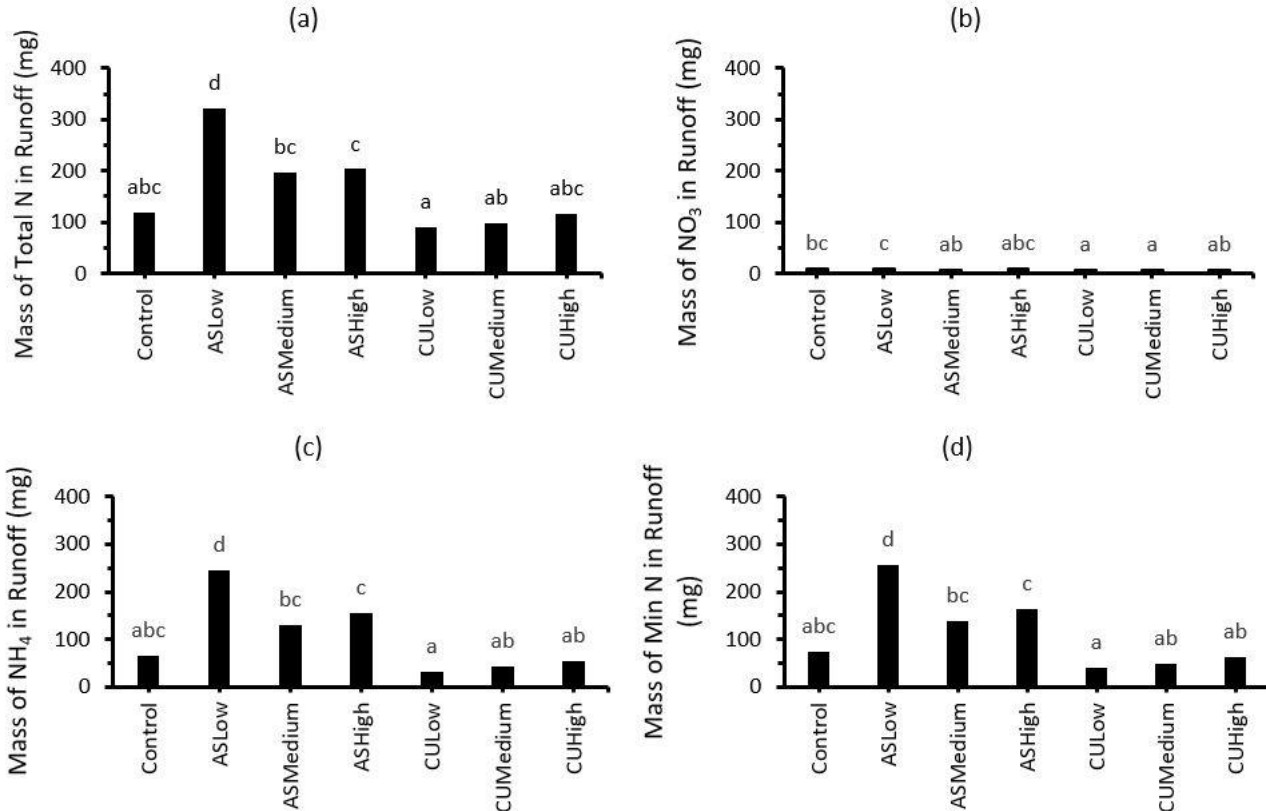

**Figure 4.** Masses of (**a**) total N, (**b**) $NO_x$, (**c**) $NH_4$ and (**d**) mineral N ($NH_4$ + $NO_x$) in runoff from the Control, ASLow, ASMed, ASHigh, CULow, CUMed and CUHigh treatments. Bars represent the mean of three replicates (n = 3). Same lowercase alphabets indicate statistically non-significant differences between treatments at $p < 0.05$.

## 4. Discussion

### 4.1. Nitrogen Release from Organic and Inorganic Sources

Leaching with dilute 0.005 M $CaCl_2$ displaced >99% of the N added as $NH_4Cl$ but only 15% of that added as CropUp[TM], with $NH_4$ being the dominant form of mineral N in the leachate. This low recovery of mineral N from CropUp[TM] results from the slow rate of nutrient release from this organic material (i.e., 0.3 d$^{-1}$) relative to readily available inorganic substrates (i.e., 0.9 d$^{-1}$). Importantly, this slow rate of release of N from CropUp[TM] would be highly beneficial for the nutrient dynamics in terms of optimising N availability relative to plant uptake requirements and reducing the N loading in the soil solution where it is susceptible to leaching and runoff loss, such as in sandy soils. Furthermore, although the leachate $NO_3$ concentrations were very low for all N substrates, CropUp[TM] tended to encourage nitrification relative to the control and $NH_4Cl$ treatments,

possibly through the introduction of nitrifying bacteria within its organic substrate, or by stimulating the resident, albeit limited, nitrifying bacteria population within the sand. Stimulating the microbial community in terms of functionality and diversity is highly beneficial in sandy soil, particularly in terms of nutrient cycling and immobilisation, which can act as a mechanism for nutrient storage and subsequent release, and for reducing leaching losses in low-cation exchange capacity (CEC) materials [33–35].

The loss of mineral N primarily as $NH_4$ demonstrates the sand's poor ability to retain this cation, thereby encouraging potential losses through leaching and runoff. The cation exchange capacity (CEC) of the sand is 1.1 cmol$_+$/kg (equivalent to 6.16 mg of negative charge per 40 g of sand expressed as N (Table 1)). The 0.005M $CaCl_2$ leaching solution provided a ready supply of Ca capable of competing with $NH_4$ for the limited cation exchange sites. Over the leaching study, approximately 0.35 mmoles of Ca was added to the sand, being equivalent to ≈10 mg of N. This mass of Ca exceeded the amount of negative charge (6.16 mg) available to retain the 9 mg of N added in the $NH_4Cl$ and CropUp$^{TM}$ treatments. Furthermore, the rapid loss of $NH_4$ from the sand during the early leaching events suggests that the sand's cation exchange sites may have a stronger preference for Ca relative to $NH_4$, thereby favouring $NH_4$ displacement and leaching with the 0.005 M $CaCl_2$ solution. Although this effect was more pronounced for the $NH_4Cl$ treatment, similar behaviour may be expected for the CropUp$^{TM}$ treatment. Therefore, the extremely low CEC of the sand coupled with a higher preference for Ca may explain the similar 2M KCl-$NH_4$ concentrations measured between treatments (i.e., 0.5 cmol$_+$/kg). The poor $NH_4$ retention by sandy soils would not only encourage N leaching, but would also result in a loss of this nutrient in runoff and lateral soil water flow (see below Rainfall runoff trial).

The mass of mineral N recovered from the CropUp$^{TM}$ and $NH_4Cl$ treatments in excess of the control was 1.04 and 9.45 mg, respectively (Table 3). Given each treatment received 9.72 mg of N, the percentage recovery of N added to the CropUp$^{TM}$ and $NH_4Cl$ treatments was approximately 15% and 100%, respectively. This suggests that nearly all of the N added as $NH_4Cl$ but only 15% of that added in CropUp$^{TM}$ was readily available over the 42-day study (Table 3). The mineralisation rates for pelleted poultry manure have been reported to be approximately 10%, 23% and 36% of initial N (total N = 2–4%) after 1, 4 and 8 weeks' incubation (25 °C), respectively [36]. These mineralisation (or N release) rates are not dissimilar to those reported in this study and show CropUp$^{TM}$ as a supplementary N source to inorganic fertiliser, which would have been most beneficial for ryegrass establishment and sustainability during the latter stage of the 42-day growth period when inorganic N sources had been depleted and the grass growth rates were high.

CropUp$^{TM}$ contains approximately 4500 and 40 mg/kg of 2M KCl extractable $NH_4$ and $NO_x$, respectively. Of the 9.72 mg total N added in 0.347 g of CropUp$^{TM}$, 1.56 mg was as $NH_4$, and 0.01 mg as $NO_x$. Therefore, approximately 8.1 mg of N was present in non-mineral forms, probably as organic N. Over the 42-day leaching study, approximately 1.44 mg of mineral N ($NH_4 + NO_x$) was recovered (>92%; Table 3), which was very similar to that added in CropUp$^{TM}$. This shows that leaching with 0.005 M $CaCl_2$ effectively removed mineral N from CropUp$^{TM}$ with negligible contributions from other fractions, such as organic matter. CropUp$^{TM}$ has a C:N ratio of 7.6 and a total N content of 3%, which may be expected to encourage the mineralisation of organic N to $NH_4$ [37]. The sand may therefore lack a significantly large functional microbial community capable of organic matter mineralisation and subsequent nitrification. Limited nitrification in the sand is supported by the absence of $NO_3$ in the $NH_4Cl$ treatment (Figure 1). However, organic inputs to sands have been shown to steadily increase organic matter and to alleviate poor microbial activity [12,19,33].

### 4.2. Rainfall Runoff Trial

The addition of CropUp$^{TM}$ (in conjunction with $(NH_4)_2SO_4$) improved plant growth, plant cover and, consequently, N uptake relative to the treatments based on $(NH_4)_2SO_4$ alone. Importantly, a higher uptake effectively removed N from the soil pore water where it

was more susceptible to loss through runoff. The reason for the lower N uptake and lower plant cover for the inorganic N treatments is unclear, but may be a consequence of: (a) the high solubility and mobility of mineral N following the solubilisation of $(NH_4)_2SO_4$ (i.e., 76.4 g/100 g water; [38] relative to CropUp$^{TM}$ (Figure 1)); (b) the possibility that, in the CULow, CUMedium and CUHigh treatments, some of the $NH_4$ released from $(NH_4)_2SO_4$ may have subsequently been retained by CropUp$^{TM}$; and/or (c) the plant was able to extract nutrients from CropUp$^{TM}$ over the growing period.

First, following dissolution, soluble $NH_4$ is highly mobile due to the very low CEC of the sand (Table 1 and Figure 1), and this cation (along with other essential nutrients added to the basal solution) may have been transported with surface-applied water deeper into the soil tray and away from the ryegrass seeds and emerging plant roots. If the availability of N and other nutrients that are critical for seedling survival was inadequate, then poor grass establishment and survival would be expected. This may explain the poor establishment growth and percent coverage of the ryegrass cover for the treatments receiving $(NH_4)_2SO_4$ alone.

Second, the inclusion of ($\approx$14.5% by weight) zeolite in CropUp$^{TM}$, coupled with slower dissolution rates, may have enabled the retention of some of the solubilised $NH_4$ closer to the point of soil incorporation, and hence closer to the developing roots. The effect of the added zeolite on $NH_4$ behaviour was not investigated specifically in this study but based on the strong preference of zeolite for $NH_4$ [39,40]. A proportion of the added N may have been preferentially adsorbed and removed from the pore water over the 42-day growth period. If the zeolite retained a proportion of the added $NH_4$ close to the source, and a proportion of this cation was subsequently available for plant uptake, this may have aided higher plant growth in the treatments that received CropUp$^{TM}$.

Third, plants have strategies to enhance nutrient uptake from the rhizosphere, such as by exuding organic acidic anions (citrate, malate and carboxylates) from plant roots [41–44], by the release of H ions from the roots to maintain cation/anion balance during nutrient uptake [42,45], or by stimulating nutrient mineralisation of the organic substrate [46–48]. The increased availability of one nutrient (e.g., P) by plant root exudates can cause a concomitant increase in the availability of co-precipitated or complexed nutrients (e.g., Mn) [43]. Although the reason for the enhanced nutrient uptake in the presence of CropUp$^{TM}$ was not investigated in this study, other researchers have reported that ryegrass (*Lolium rigidum* and *Lolium perenne* L.) roots can encourage the dissolution of phosphate and other nutrients from sparingly soluble inorganic and organic soil fractions roots due to proton excretion [44,45].

Given the slow release of N from CropUp$^{TM}$ (Figure 1), much of the mineral N measured in the runoff from the $(NH_4)_2SO_4$ + CropUp$^{TM}$ treatments most likely originated from the inorganic $(NH_4)_2SO_4$ component, which had not been taken up by the ryegrass during the growth period. Furthermore, the loss from the treatments receiving CropUp$^{TM}$ was not significantly different from that from the control, suggesting that N had been released from CropUp$^{TM}$ at a rate that matched the plant N uptake requirements. This match between N availability and plant uptake is supported by the significantly ($p < 0.05$) higher N content of the ryegrass grown in the presence of CropUp$^{TM}$ (Figure 3). Therefore, the slower release rate of N in the presence of an established plant cover clearly demonstrates the beneficial effect of CropUp$^{TM}$ on increasing plant N uptake and reducing N runoff losses compared with inorganic N fertiliser alone. The higher plant uptake of macronutrients and micronutrients demonstrates that CropUp$^{TM}$ may also act as a slow-release fertiliser.

The nitrogen use efficiency (NUE) was higher in the treatments incorporating CropUp$^{TM}$, demonstrating the benefits of organic materials in N fertiliser management. The NUE values for the ryegrass that received inorganic N fertiliser only are similar to the NUE reported for the urea N applied to the ryegrass, being 33–47% for N rates of 17–50 kg/ha [49]. The recovery of fertiliser N as determined from the NUE of major agricultural systems in Australia ranges from 28% in sugarcane to 45–62% for irrigated cotton and 78% for dairy [30], compared with an estimated global NUE of 33% [32]. Clearly, the inclusion of

organic materials as an N source can be highly beneficial for increasing NUE in pasture and cropping systems, and represents a key strategy to better manage N in the environment with the intensification of agriculture [31,32,50,51]. Importantly, the inclusion of organic materials capable of supplying plant nutrients may represent one step in the development of cropping systems on marginal soils such as sands.

Our findings extend previous research on the mechanisms involved in the beneficial effects of combined organic and inorganic fertilisers. Several studies have shown that the addition of organic fertiliser to inorganic fertiliser has a positive effect on soil chemical and biological properties [52,53]. Wen et al. (2016) [54] showed that this fertilisation regime increases nutrient uptake via stimulating root growth. It also reduces N leaching and enhances denitrifier activity [55]. Here, we provide further evidence that this combination benefits plant N use efficiency via reduced N runoff losses. This has important implications particularly for increased NUE of farming systems in high rainfall geographical locations.

**5. Conclusions**

The combination of inorganic and organic N substrates reduced the total and mineral N runoff losses compared with inorganic N alone in coarse-textured sand. The higher N uptake by the plant cover provides strong evidence that the combined use of inorganic and organic sources has the potential for increasing NUE by reducing fertiliser losses in surface runoff and synchronising better the N availability with plant requirements. Inorganic N fertiliser can be applied at low rates to provide sufficient N at the early growth stage, with organic substrates, such as composted manures, supplying N at later stages, particularly when the plant growth rate is high.

Furthermore, the potentially slower release of N from organic materials such as CropUp$^{TM}$ may reduce the impacts of other loss pathways (e.g., leaching and sub-surface lateral flow) relative to highly soluble inorganic N fertilisers, particularly when the release rate better matches plant N uptake requirements. These organic materials have widespread applications to sandy soils not only in agriculture, but also in other activities that involve a high N application to sand, such as golf courses and sporting ovals.

**Author Contributions:** Conceptualisation, I.P., C.P.-L. and M.R.; methodology, I.P., M.R. and D.M.; formal analysis, I.P. and D.M.; investigation, I.P.; resources, C.P.-L., I.T., B.S. and B.H.; data curation, I.P.; writing—original draft preparation, I.P.; writing—review and editing, I.P., C.P.-L., M.R., I.T., B.S., D.M. and B.H.; supervision, M.R.; funding acquisition, C.P.-L. All authors have read and agreed to the published version of the manuscript.

**Funding:** This research was funded by the Australian Government's Cooperative Research Centres Projects, grant number CRCPFIVE000015.

**Institutional Review Board Statement:** Not applicable.

**Informed Consent Statement:** Not applicable.

**Data Availability Statement:** Not applicable.

**Acknowledgments:** We would like to thank the Cooperative Research Centres Projects (grant number CRCPFIVE000015) for funding this project.

**Conflicts of Interest:** The authors declare no conflict of interest.

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
