# Peer review of "Combination of Inorganic Nitrogen and Organic Soil Amendment Improves Nitrogen Use Efficiency While Reducing Nitrogen Runoff"

_nitrogen, doi:10.3390/nitrogen3010004_

Round 1
Reviewer 1 Report
The introduction is not related to the actual experience. Long sentences with wrong syntax. Drawings made carelessly, each different. The description of the experience is chaotic, incomprehensible. Equipment schematics would be handy. Very short time of the experiment. The conditions are very far from the real ones. Lots of editorial errors. Badly spelled units. I marked some of them in the comments. Why such methodology of chemical analyzes and why such extracts. There are no standards in Australia? The discussion is largely a repetition of the results. Very general and obvious conclusions without research. The work looks as if a fragment was taken out of the whole.

Author Response
We thanks the reviewer for all the constructive comments, which are very helpful in guiding us to improve the quality of this manuscript during the revision. We have addressed all issues raised and revised the manuscript carefully.
- The introduction is of relevance to the experiments. It introduces the importance of using organic fertilisers to improve nitrogen use efficiency. Related previous investigations were identified, and subsequently, the proposed experimentation was described to fill some of the research gaps.
- We have now consulted the editing services of MDPI to check the English.
- Equipment schematics are presented in the cited reference (25. Humphry, J.; Daniel, T.; Edwards, D.; Sharpley, A. A Portable Rainfall Simulator for Plot-Scale Runoff Studies. Appl. Eng. Agric. 2002, 18, doi:10.13031/2013.7789.).
- Ryegrass is one of the fast-growing pasture species. The two experimentations were set up for 42 days each, a length of time that allowed clear differences between treatments to be revealed. Additionally, other studies have used similar timeframes; therefore, our results can be comparable.
- The conditions for the experiments were specially designed by Humphry et al. (2002) for this type of study.
- The methodology of extractions used in this study followed the standard soil and chemical protocol of Rayment and Lyons (2010), which is a standard used in Australasia and beyond. Soil Chemical Methods - Australasia, George E Rayment, David J Lyons, 9780643101364 (csiro.au). https://www.publish.csiro.au/book/6418
- We have added an additional discussion to emphasise the importance of our findings and link those to previous investigations. Lines 477-485.
“Our findings extend previous research on the mechanisms involved in the benefi-cial effects of combined organic and inorganic fertilisers. Several studies have shown that the addition of organic fertiliser to inorganic fertiliser has a positive effect on soil chemical and biological properties [53,54]. Wen et al. (2016) [55] have shown that this fertilisation regime increases nutrient uptake via stimulating root growth. It also re-duces N leaching and enhances denitrifier activity [56]. Here, we provide further evi-dence that this combination benefits plant N use efficiency via reduced N runoff losses. This has important implications particularly for increased NUE of farming systems in high rainfall geographical locations.”

Reviewer 2 Report
Consider in much more depth in the Discussion the uniqueness of the findings and what they contribute to the knowledge base behind increasing use-efficiency or optimizing yield and inputs. The paper must show how it relates to similar work in other parts of the world.
Line 293-Figure 2 should be reconstructed for a better understanding of the presented results. That means that treatments should be grouped with respect to N forms.
Line 341-Table 4. Too many parameters which are not related to the main topic of the paper. Also, you didn't refer to it in the text except for NUE, so my suggestion is to simplify the table design excluding excessive soil parameters.
Author Response
Consider in much more depth in the Discussion the uniqueness of the findings and what they contribute to the knowledge base behind increasing use-efficiency or optimizing yield and inputs. The paper must show how it relates to similar work in other parts of the world.
>> More discussion is added in lines 477-485
“Our findings extend previous research on the mechanisms involved in the benefi-cial effects of combined organic and inorganic fertilisers. Several studies have shown that the addition of organic fertiliser to inorganic fertiliser has a positive effect on soil chemical and biological properties [53,54]. Wen et al. (2016) [55] have shown that this fertilisation regime increases nutrient uptake via stimulating root growth. It also re-duces N leaching and enhances denitrifier activity [56]. Here, we provide further evi-dence that this combination benefits plant N use efficiency via reduced N runoff losses. This has important implications particularly for increased NUE of farming systems in high rainfall geographical locations.”
References:
- Goyal, S.; Chander, K.; Mundra, M.C.; Kapoor, K.K. Influence of Inorganic Fertilizers and Organic Amendments on Soil Organic Matter and Soil Microbial Properties under Tropical Conditions. Biol. Fertil. Soils 1999, 29, 196–200, doi:10.1007/s003740050544.
- Kaur, K.; Kapoor, K.K.; Gupta, A.P. Impact of Organic Manures with and without Mineral Fertilizers on Soil Chemical and Biological Properties under Tropical Conditions. J. Plant Nutr. Soil Sci. 2005, 168, 117–122, doi:10.1002/jpln.200421442.
- Wen, Z.; Shen, J.; Blackwell, M.; Li, H.; Zhao, B.; Yuan, H. Combined Applications of Nitrogen and Phosphorus Fertilizers with Manure Increase Maize Yield and Nutrient Uptake via Stimulating Root Growth in a Long-Term Experiment. Pedosphere 2016, 26, 62–73, doi:10.1016/S1002-0160(15)60023-6.
- Kramer, S.B.; Reganold, J.P.; Glover, J.D.; Bohannan, B.J.M.; Mooney, H.A. Reduced Nitrate Leaching and Enhanced Denitrifier Activity and Efficiency in Organically Fertilized Soils. Proc. Natl. Acad. Sci. 2006, 103, 4522–4527, doi:10.1073/pnas.0600359103.
Line 293-Figure 2 should be reconstructed for a better understanding of the presented results. That means that treatments should be grouped with respect to N forms.
>> We thank the reviewer for pointing this out – we have regrouped the results according to the N forms.
Line 341-Table 4. Too many parameters which are not related to the main topic of the paper. Also, you didn't refer to it in the text except for NUE, so my suggestion is to simplify the table design excluding excessive soil parameters.
>> We propose to keep all macro-and micro-nutrient parameters in Table 4 as those data are referred to in the text (lines 321-325).

Reviewer 3 Report
I read the draft by Phillips et al. “Combination of inorganic nitrogen and organic soil amendment improves nitrogen use efficiency while reducing nitrogen runoff”. This work represents an agricultural technology analysis aimed at improving sand soil nitrogen use efficiency, combining inorganic and organic fertilization. The text is perfectly clear in meaning and the experimentation well-conceived. The results, though simple in essence, suggest that the combination of inorganic and organic nitrogen substrates do help in reducing total and mineral rainfall runoff losses of nitrogen as compared to inorganic sources alone. I believe this work will be useful for the community and I’m happy to recommend publication in its present form. My only suggestion would be to better clarify, at the beginning of the results section, what the experimental set up was about. As it stands, the texts jumps right on an actual observation, without explaining what is being tested or what the controls and treatments were. Although it may become obvious from the materials and methods, most readers may not read them, and will not be obvious to non-experts.
Author Response
We thank the Referee for his/her positive assessment. The suggestion on improving the clarity of the result from the beginning is very helpful in guiding us to increase the quality of this manuscript. In response to this suggestion, we have now added new sentences at lines 248-256.
“A rainfall simulation trial was undertaken to evaluate the impacts of combined inor-ganic and organic fertiliser on nitrogen runoff losses compared with conventional in-organic fertiliser ((NH4)2SO4) alone. A nitrogen release experiment was concurrently undertaken to investigate N mineralisation kinetics.
3.1 Nitrogen Release from Organic and Inorganic Sources
A leaching experiment was undertaken to investigate the N mineralisation kinetics in the coarse-textured sand used in the rainfall simulation trial. The results showed that negligible NH4 and NOx were leached over the 42-day period in the control treatment (no N applied), confirming the very low mineral N status of the sand (Figure 1)...”
Round 2
Reviewer 1 Report
Units notation corrected everywhere. Not all comments were taken into account, and no explanations from the authors. Editing errors previously marked have not been corrected.
Author Response
We have now addressed all remaining comments (please see attached).
